# Aging in Place with Age-Related Cognitive Changes: The Impact of Caregiving Support and Finances

**Alexandra Wagner** 

Department of Occupational Therapy, Yeshiva University, New York, NY 10033, USA; Alexandra.wagner@yu.edu

**Abstract:** In the United States, aging in place is a common concept that refers to older adults' desire to remain in their homes as they age. However, this ability to age in place is a complex process that is not only impacted by the home's accessibility or individual functional abilities. This paper aims to examine different factors, such as home environment and home modification, caregivers, finances, and other supports present in the participants' lives, that impact older adults with age-related cognitive changes (ARCC) (in)ability to age in place. Qualitative interviews with older adults with ARCC (*n* = 5) and their caregivers (*n* = 5) were conducted. The participants' experiences while aging in place indicate that finances and caregiving support greatly impacted their lives at home and ability to age in place. Personal finances dictated where some of the participants could age and the support, they could afford from home health aides. Additionally, informal and formal caregivers were an important source of support that aided in the older adults' ability to remain home. As researchers, we need to continue to address personal finances and the support that the individual has in their lives to most effectively promote aging in place and their life at home.

**Keywords:** aging in place; older adults; dementia; cognitive changes; home



## 1. Introduction

In the United States, the term aging in place commonly refers to older adult's desire to remain in their home instead of transitioning to an institutional setting as they grow older. The overall goal of the knowledge produced from this literature is to increase the likelihood that older adults will remain in their homes, in the community, safely [1,2].

To age in place with advanced cognitive decline, beyond what is seen in typical aging adults, can create further complications and difficulties to age in place. People with Alzheimer's disease or dementia are at a higher rate for nursing home admission compared to older adults who do not have Alzheimer's disease or dementia [3]. 50.4% of individuals in institutional settings have a diagnosis of Alzheimer's or other age-related cognitive change diagnoses [4]. This statistic further indicates the difficulties this population faces to be able to age in place. In this paper, this group of aging adults is referred to as older adults with age-related cognitive changes (ARCC). The term includes those diagnosed with dementia or Alzheimer's disease and those with cognitive changes that are more advanced than what is typically seen in aging adults that impact their daily lives.

Caregivers, informal or formal, provide the necessary support to older adults with ARCC to remain in their homes. For all older adults in general, 83% of the care these individuals receive is provided by an informal caregiver. Informal caregiving is highly gendered, and daughters are more likely than sons to provide informal care [5]. It was reported that older adults with ARCC who are aging in place require over 100 h of care a month [6]. Caregivers, formal or informal, assist with basic care needs, home management, and medication management. They ensure the older adult is safe while in the home. Additionally, caregivers support the coordination of medical care and utilization of medical services. Research demonstrated that older adults with ARCC with the support of a caregiver utilized health services and had increased less difficulty with coordination of care

compared to those who live alone [7,8]. The role that caregivers play in the lives of older adults with ARCC is vital, caregiver stress and burden increases placement of nursing home admission, further supporting the necessity that caregivers play to promote aging in place [9,10].

A large subsection of the aging in place literature focuses on home modification to promote accessibility throughout the home. Safety concerns are often the main priority when performing home modifications [11–13]. Modifications to the built environment increase adults' independence and allow for routines to continue to be performed [14]. Home modification limit the decline in performance of activities of daily living and instrumental activities of daily living [15]. However, there is a balance to the implementation of modifications; certain modifications can disrupt an individual's with ARCC physical space too much, especially if the home modifications are not chosen by the person who lives in the house [14]. Although physical modifications can increase accessibility, they can also confuse the individual with ARCC and irritate them. The increase in confusion and irritation is a fear that family members of those with ARCC have if they modify their homes [16]. Home modifications should support an individual's physical and cognitive needs. Unfortunately, home modifications to support a person's cognitive needs are not made at the same rate that modifications are made to compensate for the physical change that occurs [12].

There are many older adults whose lack of financial instability threatens their ability to remain in their homes [17]. Often lower-income older adults' households are unhealthy environments and cause adverse health effects because of poor housing conditions, including injuries, respiratory infections, lead poisoning, and other chronic conditions [18]. The current poverty level for adults in the United States aged 65 and older is 9.7% [19]. However, not included in the poverty measure are out-of-pocket health costs and other measures including but not limited to capital losses and gains and income and payroll taxes. With the inclusion of additional measures that are not uncommon in older adults' lives, poverty rates are 57–89% higher than the official poverty rate [20]. In other words, as people age, the additional costs associated with the aging process impact the financial stability of older adults. If they are included, it increases the number of older adults living in poverty dramatically.

There is an assumption that older adults have limited housing-related expenses because they live in homes paid off. However, in the United States, one in three older adult households experience the burden of high housing costs [21]. In 2015, almost 44% of older adult households spent 1/3rd or more of their income on housing costs [22]. Some older adults live on a fixed income, making it financially difficult to afford a home when housing prices are constantly rising [23]. There is often a choice that needs to be made regarding individuals' use of financial resources [24].

Although there is an extensive amount of research that exists about aging in place, there is no one size fits all answer to aging in place or where an older adult should live [25,26]. However, someone's (in)ability to age in place is dependent on a multitude of different factors. Difficulties older adults with ARCC face while aging in place include safety-related problems, an inability to complete ADLs, medicine management, caregiver stress, and a lack of formal care [27]. Compounded, these difficulties, without intervention, can create an almost impossible situation for aging in place. To most effectively promote aging in place for this population, the difficulties and differences that exist in each person's life need to be explored from their perspective directly and addressed.

The aim of this research was to document the experiences older adults with ARCC faced while aging in place to document the aspects in their life that allowed them (or made it difficult) to age in place. Secondly, this paper examined how the factors that they identified that supported or did not support their ability to age in place interacted with one another. These supports include the home environment and home modification, caregivers, finances, and other supports present in the participants' lives. It is expected that the older adults who have more supports in their life can age in place safer and remain in their

home longer than those who do not have the necessary supports. The examination of this population of older adults' lives is important because older adults' ability to remain in their home as they age is dependent on many factors that interact. How different factors interact need to be further explored to advance the field and more effectively promote aging in place.

## 2. Materials and Method

### 2.1. Participants and Recruitment

Participants were recruited from organizations that support individuals with ARCC and their families. Staff members in the organization shared basic information about the study and the researcher's contact information to their members who fit the inclusion and exclusion criteria. To participate in the study, an individual needed to be at least 60-years old, living in a home within the community. Participants could not live in an assisted-living facility, independent living facility, group home, nursing home, or institutional setting.

Additionally, participants had to be diagnosed (or be suspected of having) dementia or Alzheimer's disease or had cognitive changes that began as they aged, which impacted their daily lives. Examples of cognitive changes that impacted their daily life included needing assistance completing activities of daily living or additional assistance to complete tasks around their home safely.

Caregivers of the participants with ARCC were encouraged to participate in the study to inform the research question and as a form of triangulation. This could include formal or informal caregivers, friends, neighbors, or individuals who support the older adults' life at home. The study sample consisted of five participants with ARCC and five caregivers. See Table 1 for demographic information of the participants.

**Table 1.** Participant Characteristics (*n* = 10).

| Name of Older Adult with Age-Related Cognitive Changes (ARCC) (*n* = 5) | Age/Gender | Race/Ethnicity | Number of Years with Cognitive Changes | Caregiver/Relation to the Older Adult with ARCC (*n* = 5) |
|---|---|---|---|---|
| Mary-Ann | 86/Female | Caucasian | 1 | Deborah (Daughter) |
| Maria | 76/Female | Caucasian | 7 | Richard (Husband) |
| Carol | 90/Female | Caucasian | Unknown | Aviva (Friend) Diane (Home Health Aide) |
| Henry | 90/Male | Caucasian | 2 | Sandy (Daughter) |
| Colleen | 64/Female | Caucasian | 1 | N/A |

### 2.2. Methods

Participants consented to interviews, diary keeping, and observation sessions to allow participants with different cognitive abilities to participate. Different methods were included in the research design to be as inclusive as possible to different participants and their different abilities. All the participants could verbally communicate their thoughts and experiences; therefore, interviews were the primary medium for data collection. Interview questions sought to understand the participants life at home and the different factors in their life that supported their life at home. Interviews with caregivers confirmed information stated by the individual with ARCC and provided additional information about the older adult with ARCC life at home.

Interviews were completed in the participant's home because their home gave the person environmental cues and contexts to the interview topics [28]. The CORTE (Consent, maximizing Responses, Telling the story, and Ending on a high) was adapted to promote the participant's inclusion and active involvement in the research [29]. Participants were interviewed multiple times to gain additional information and not overwhelm the participants with one long interview session. However, retention of participants was difficult. Two of the participants with ARCC passed away, one moved to a nursing home, and

another participant had recurrent falls, which resulted in hospitalizations and long-term stays in rehabilitation centers.

*2.3. Data Analysis*

The qualitative software, NVivo, was used to analyze the data. The analysis began with the open coding stage, where general codes are assigned to the data [30,31]. Each participant's data was looked at, and the items in their life that impacted their ability to age in place were coded. After each participant's data was coded, these codes were brought together to create larger categories representing the participants' important experiences and adequately worked towards answering the research question, axial coding [30,32]. Throughout this iterative process also comes selective coding, where codes are re-conceptualized and made to tell a story of the data collected [31,33].

*2.4. Study Approval*

The Institutional Review Board at SUNY Stony Brook University (protocol code 1181693, date of approval 18 May 2018) approved this study protocol. Participation in this study was voluntary, and all identifying details were changed to protect the participants' identity. Written informed consent was obtained from the participants to publish this paper. The participants with ARCC needed to pass a capacity-to-consent assessment before signing the consent form. If the individual was unable to pass the capacity-to-consent form, their caregiver signed for them.

## 3. Results

The participants' revealed that the main aspects in their life that contributed to their ability to age in place were caregiving support and finances. The participants' finances were brought up in terms of the home they could afford to live in and affording to hire a formal caregiver. The support provided by caregivers was a necessity in the participants lives. However, these two just mentioned, findings worked together, and sometimes against each other, to foster the participants' ability to age in place. Despite it being a large subset of the literature within their field of study, home modifications were not found to significantly impact any participants' lives or ability to remain in their homes.

*3.1. Finances*

3.1.1. Housing

The participants discussed their financial situation as promoting their ability to remain in their home or threatening their ability to remain in their home. Some of the participants financial situation dictated where they were able to age in place. For the others, their financial security ensured that they could remain in the home they loved. The examples from the participants Mary-Ann and Colleen highlight the impact of finances on the place they could live in based on their financial situation.

Mary-Ann, at the time of her involvement in the study, had been living for the past two years in an affordable housing apartment complex. When she was asked if she could see herself staying in the apartment moving forward, she was unsure. She said, "I can't say if I've got the money tomorrow." Her daughter added that one of the reasons she liked her mother's apartment was the affordability of it. Mary-Ann moved out of her two previous homes because of her inability to afford them. Her first home she moved out of was a large home but it was too big for her, and "[she] needed the money [she] could get out of it." The second place she moved out of was an apartment that was move expensive than her current residence. Mary-Ann spoke with fonder memories of her old apartment than she did her current residence. She liked the apartment she was living in; however, she explained, "It's not like let's see, I lived in [a different complex] for a while- rented there. But it's a whole different world there, they have money. They have retired, and their husband died, and I had a lot of friends that I knew before that lived there. So, I rented for a while."

Similarly, Colleen was concerned about her financial status and her ability to remain in her home. Colleen lived in affordable housing with two roommates. She paid for her rent with her Social Security Disability Income (SSDI) payments. However, the possibility that her SSDI would be cut and the inability to pay rent in the future was a source of stress in her life. The possibility of not being able to pay her rent worried Colleen so much that she felt that moving to a nursing home would be better for her. She explained, "I don't know what's gonna happen. I won't even make the rent with what they cut me, you know?" She went on to say, "so I was just praying I could go into a nursing home." Both Mary-Ann's and Colleen's financial status impacts the home an older adult lived in now and will be able to live in as they continue to age.

### 3.1.2. Affording Formal Caregiving

The participants' financial status also impacted the formal caregiving they could (or could not) have in the home. This was illustrated best in the lives of Carol and Henry. Both of the participants had home health aides (HHAs) in their home supporting their daily life. The support provided by the HHA was necessary to ensure that they could remain at home safely.

According to Carol's close friend, Aviva, Carol would not be able to live in her home without having a 24 h HHA living with her. Carol had experienced numerous falls and her son decided it was unsafe for her to be living alone. Her finances afforded her the ability to hire and pay, out of pocket, for a 24 h aide to live with her.

Unlike Carol, Henry and his daughter Sally did not have the financial means to pay for additional help; they needed to apply for Medicaid to receive assistance from a HHA. Sally said it would be impossible to pay for the HHAs out of pocket. Sally took care of all of Henry's finances. Detailed extensively by her, the Medicaid application process was complicated, and she faced barriers along the way. To begin applying for Medicaid, Sally started with collecting statements to prove her father's financial status. The process to be approved for Medicaid took eight months, even with help from an elder attorney and Henry's other adult daughter.

The personal finances of the older adults impacted their ability to hire necessary caregiving support. Carol's ability to pay out of pocket for a HHA was simpler compared to Henry and his daughter's experience applying for Medicaid.

### *3.2. Caregiving Support*

Caregivers, informal or formal, were a key support in the lives of the participants. All five of the participants had caregiving support in some capacity. However, the amount of support that the family or formal caregivers provided varied. Family members supported the individual with ARCC with managing finances, medication management, grocery shopping, activities of daily living, and basic tasks around the home. All of the activities that the informal caregivers provided was to ensure that the older adult was safe in their home. For some of the participants, the support provided to them made it possible to live safely at home. If it were not for the older adults' caregivers, they would not be able to remain in their homes.

The support and assistance of a caregiver, informal or formal, was important for the older adults in this study and was highlighted through Colleen's experiences at home. Colleen had two adult children, one who lived in a different state and helped her with tasks like laundry and ensured she had food in her fridge, and one who lived locally but did not visit frequently. Colleen would have preferred to live with her daughter; she explained," [my daughter] thinks I'm not um-you know she thinks I'm too much." Colleen's current level of need would be too difficult for her daughter, who worked full time and had a newborn baby, to provide the necessary support. While interviewing Colleen, she expressed a desire to move to a nursing home. Colleen accepted the likelihood that she would not be able to maintain her life at home and accepted the possibility that she would need to move to a nursing home.

## 4. Discussion

The aim of this research was to understand the different aspects within the lives of five older adults with ARCC that supported or created barriers to aging in place. Older adults desire to remain in their homes [34]. Older adults with ARCC value being able to remain in their home because of the security their home affords them [35,36]. For some older adults, leaving their homes is unimaginable [37]. The participants in this study echoed this sentiment. However, only if they had the financial means to remain in their home. For some of the participants, the living expenses and housing unaffordability impacted where they could live. After interviewing the older adults with ARCC and their caregivers, the participants revealed that finances and caregiving greatly impacted their life at home.

When researching aging in place, the older adults' finances must be considered because one in three older adults experience a financial burden due to high housing costs [21]. This was true for the participants in this study, two of the participants lived in a rented house/apartment, and their financial situation dictated where they could live. Colleen's affinity towards their home was not enough to support her ability to remain at home. In other words, despite her preference to live in her apartment in the community, her limited finances and her declining functional abilities did not support her life at home.

Colleen's experience paralleled other low-income older adults' experiences. As the cost of housing increases, a larger percentage of older adults' fixed income goes towards housing costs, making it more difficult to afford to live there [23]. Research found that aging in place might not be possible for low-income seniors [17]. Low-income older adults have different needs to age in place compared to older adults who are financially secure. Additionally, the reported high costs of health care expenditures of living with dementia compared to other disease in older adulthood further supports the need to address the financial security of this population [38]. With the known difficulties low-income seniors face as they age in place, an older adult's financial situation should not be neglected in aging in place research.

The second area that was found to impact the participants' life at home and ability to age in place was the support provided by caregivers (informal and formal). Caregivers are vital to support older adults with ARCC life at home as it was found that older adults who live alone have more unmet care needs and additional difficulties compared to those who live with their caregiver [8,39,40]. The older adults in this sample relied heavily on their loved ones and formal caregivers to support their life at home. The importance and necessity of informal caregivers in the lives of the participants in this study aligns with previous research [41].

However, the ability to hire a formal caregiver, a HHA, was impacted by the older adults' financial means. Researchers found that an aide in the home acts as a facilitator to aging in place because this population commonly requires additional care in the home [41]. It was not surprising to find that the presence of HHAs was dictated by the participants' ability to pay for their services. On average, the cost of hiring a HHA is about $12 an hour [42]. This amount of money can become burdensome for the person paying for the services, especially given the amount of care a person with ARCC can require. A second way older adults have to afford the cost of a HHA is through Medicaid.

Although older adults age 65 and older receive Medicare, it does not cover the cost of HHAs. Medicaid is a federal and state program that provides health coverage for those who qualify for it across the United States. Medicaid will pay, if a person qualifies, for caregiving support from HHAs, as well as other medical services [43]. Medicaid requirements vary by state. The Medicaid application is difficult, and people often experience difficulties throughout the process. People with more health problems and less education were found to have greater difficulty with the application and faced more barriers than those with more education and fewer health problems [44]. The participants, Henry and his daughter, were one example of the difficult and lengthy process people face when applying for Medicaid.

All the factors that impact an older adult with ARCC are linked together in complex ways. As researchers, we need to be aware of the variety of connecting factors in their

lives to effectively understand their experiences and promote aging in place in meaningful and lasting ways that benefit aging adults across various socioeconomic factors. Although, when someone ages in place, they are doing so in their home, it does not mean that the discussion surrounding aging in place needs to focus on functional abilities and the house they live in, in isolation. This study pointed to key areas, finances and caregiving support, in older adults' lives that impacts aging in place and the ability to do so.

## 5. Conclusions and Future Directions

Although aging in place, on paper, presents as a simple concept, it should not be simplistically understood or applied to a person's life. One reason for this is the complexity that exists in a person's life. The financial situation of older adults with ARCC had great impacts on their (in)ability to age in place. It also impacted the support that they could or could not afford. The support of informal caregivers was paramount for the lives of older adults with ARCC. Aging in place is more than just an older adult and the home they live in. Future research should examine the just mentioned aspects of older adults as well as other aspects of older adults lives that contribute to older adults (in)ability to age in place.

One area, related to this topic, that could be explored further is health literacy. Health literacy is important for this population as older adults often have chronic conditions and other age-related functional losses that require a certain level of health literacy [45]. Research based on a nationally representative study found that adults with the lowest average health literacy were those aged 65 years old and above. When considering older adults with cognitive changes, cognitive changes further create challenges while navigating the health care system. Research found significant results for the association of impaired verbal fluency and memory and inadequate health literacy [46]. Older adults with ARCC are at higher rates of difficulty understanding and then acting upon their health condition based on the advice given to them by medical professionals. The challenges to understand the health care system or processes like applying for Medicaid impacts the services that older adults with ARCC can access such as caregiving services/support. Thus, impacting older adults' ability to age in place.

Finally, future research should continue to explore this area amongst racially diverse aging adults and investigate how intersections of race impact the factors that impact aging in place that were identified in this study. African Americans over the age of 65, compared to other racial groups in this age group, face higher rates of poverty [19,47]. Higher rates of poverty impact the ability to age in place [17]. Previous studies identified how racial minorities are less likely to stay in their current residence compared to Caucasian seniors [48]. However, additional research is needed to further support different groups of aging adults based on their differences and identified needs.

**Funding:** This research received no external funding.

**Institutional Review Board Statement:** The study was conducted according to the guidelines of the Declaration of Helsinki and approved by the Institutional Review Board of SUNY Stony Brook University (protocol code 1181693 and 18 May 2018).

**Informed Consent Statement:** Informed consent was obtained from all subjects involved in the study.

**Conflicts of Interest:** The author declares no conflict of interest.

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
