# Peer review of "Aging in Place with Age-Related Cognitive Changes: The Impact of Caregiving Support and Finances"

_societies, doi:10.3390/soc11020031_

Round 1

Reviewer 1 Report

This paper has a great potential but lacks of method and the basis is too small to be able to make a general discussion about the topic. I always suggest to set up a IMRAD style of managing ideas but in this case I would suggest to make it balanced. Five cases are few and it is needed a well organized survey in order to be able to be guided at the moment of the evaluation of the data. So I would suggest to start with a well organized list of issues to be solved (state of the art), aim of the topic, strategies in the research management, selection criteria (database, interviews etc...), expected results, future development. The potential is high but it is needed more work in method and research implementation.

Reviewer 2 Report

Interesting topic. Well-written. Interesting research group.

TITLE

Is aging complex? The study outcomes suggest that just two important factors determine optimal aging in place: finance and caregiving support. There are many ways of choosing a Title, and one is to reflect the fundamental Research Question.

ABSTRACT

The Abstract talks of exploring a topic; this is a field with many studies, so it should be possible to formulate clear hypotheses and research questions.

Interesting idea to interview both older adults and their caregivers.

INTRODUCTION

Well-written Introduction. It covers the relevant facts.

  1. 81 What is the difference between “To truly promote aging in place” and “To promote aging in place”; what is the function of the word ‘truly’? Is it metaphysical?

The Abstract talks of exploring a topic; this is a field with many studies, so it should be possible to formulate clear hypotheses and research questions.

“Second, this paper pushes back on the oversimplified notion that aging in place is simply a choice.” I was surprised by the authors writing this in the paper section that is meant for Research Questions. It seems the authors already know what to do without carrying out the research. The present paper is not a policy paper, it is a research paper. Please make a clear choice of what this paper is: a policy paper or a research paper.

MATERIAL AND METHODS

A good idea to contribute a qualitative study to a field with many quantitative studies. Interesting to see whether the outcomes from interviews are in line with the findings from questionnaires.

  1. 95 ‘assisted-living facility’; check the use of hyphens throughout the text.

The sample of the elderly is quite small, but the data collection was thorough.

It would be good to supply additional information on the data processing so that other researchers are able to repeat the process. Are the primary data made available to other researchers?

RESULTS

The data are professionally analyzed.

DISCUSSION

The outcomes of the study are in line with what was known from the rest of the literature. It is good to have a study with data from interviews supporting the qualitative literature.

Various topics are discussed. Good. Maybe the other reviewers want to see additional issues being discussed.

Round 2

Reviewer 1 Report

My concerns are still on the basis of the statistical study you performed. For me the number of interviews is still too small. You should find the way to organize it in a more scientific way, adding some more experiments and enhancing the scientific level of the paper. For example:

line 34-42 not clear what is the source of data

line 255-261 for example it is not a scientific sentence

Reviewer 2 Report

The authors did a good job processing some of my feedback, but other parts of my feedback still need to be processed correctly.

TITLE

There are many ways of choosing a Title, and one is to reflect the fundamental Research Question. The title now reflects an unclear Research Question. Usually, in a Research Question, a hypothesis is tested.

For example, when giving underfed children multivitamins, we predict that their scores will go up .3 SD.

Or, when patients with Alzheimer’s are studied, we expect that they complain most about factors a, b, and c.

“The impact and interaction of different factors” ….. Very unclear and very unspecific. Which factors? What does the literature tell you? What does the literature suggest you? How big will the effects be? Interaction of which factors?

There is already an extensive literature, so I expect clear hypotheses.

Now it reads more like a fishing expedition.

ABSTRACT

The Abstract talks of exploring a topic; this is a field with many studies, so it should be possible to formulate clear hypotheses and research questions.

INTRODUCTION

The Abstract talks of exploring a topic; this is a field with many studies, so it should be possible to formulate clear hypotheses and research questions.

Round 3

Reviewer 1 Report

I think the structure of the paper cannot be changed extensively due to conditions and I consider the paper now eligible for publication.

Reviewer 2 Report

The authors did a good job processing some of my feedback, but other parts of my feedback still need to be processed correctly.

TITLE

There are many ways of choosing a Title, and one is to reflect the fundamental Research Question. The title now reflects an unclear Research Question. Usually, in a Research Question, a hypothesis is tested.

For example, when giving underfed children multivitamins, we predict that their scores will go up .3 SD.

Or, when patients with Alzheimer’s are studied, we expect that they complain most about factors a, b, and c.

“The impact and interaction of different factors” ….. Very unclear and very unspecific. Which factors? What does the literature tell you? What does the literature suggest you? How big will the effects be? Interaction of which factors?

There is already an extensive literature, so I expect clear hypotheses.

Now it reads more like a fishing expedition.

ABSTRACT

The Abstract talks of exploring a topic; this is a field with many studies, so it should be possible to formulate clear hypotheses and research questions.

INTRODUCTION

The Abstract talks of exploring a topic; this is a field with many studies, so it should be possible to formulate clear hypotheses and research questions.